# Transgressing Boundaries between Community Learning and Higher Education: Levers and Barriers

**Thomas Macintyre** [1,2,*], **Martha Chaves** [2], **Tatiana Monroy** [2,3], **Margarita O. Zethelius** [4], **Tania Villarreal** [5], **Valentina C. Tassone** [1] and **Arjen E. J. Wals** [1]

1   Education and Learning Science Group, Wageningen University, 6706 KN Wageningen, The Netherlands; valentina.tassone@wur.nl (V.C.T.); Arjen.Wals@wur.nl (A.E.J.W.)
2   Fundación Mentes en Transición, Filandia 634007, Colombia; marthacecilia.chaves@gmail.com (M.C.); wayracolombia@gmail.com (T.M.)
3   Aldeafeliz ecovillage, San Francisco 253607, Colombia
4   Alianzas para la Abundancia, Las Islas del Rosario 130019, Colombia; margaritaozethelius@gmail.com
5   Corporación Nuh Jay, Pasto 520001, Colombia; tania_laureluna@hotmail.com
*   Correspondence: thomas.macintyre@gmail.com; Tel.: +31-3188343114

**Abstract:** In times of global systemic dysfunction, there is an increasing need to bridge higher education with community-based learning environments so as to generate locally relevant responses towards sustainability challenges. This can be achieved by creating and supporting so-called learning ecologies that blend informal community-based forms of learning with more formal learning found in higher education environments. The objective of this paper is to explore the levers and barriers for connecting the above forms of learning through the theory and practice of an educational approach that fully engages the heart (feelings), head (thinking), and hands (doing). First, we present the development of an educational approach called Koru, based on a methodology of transgressive action research. Second, we critically analyze how this approach was put into practice through a community-learning course on responsible tourism held in Colombia. Results show that ICT, relations to place, and intercultural communication acted as levers toward bridging forms of learning between participants, but addressing underlying power structures between participants need more attention for educational boundaries to be genuinely transgressed.

**Keywords:** transgressive learning; decolonizing pedagogies; learning ecologies; Koru Educational approach; Colombia

## 1. Introduction

It is November 2018, and Colombian students at public universities are two months into a national strike. Wages for teachers have not increased in the last 10 years, with ever diminishing resources for public universities. As a sign of demonstration, the front gate of the University of Quindío is blockaded with chairs and desks. I am forced to enter the University through a narrow passageway further down the entrance, where security guards and students are milling around. I make my way to the cafeteria where a group of students are waiting for me. As representatives from academia, the students have responded to the invitation to take part in a community-learning course on responsible tourism. As we sip coffee, and the students enter the virtual learning platform for the course, I feel the excitement begin to rise. Together with other actors we will be addressing the urgent need to bring together different forms of knowledge and practice in addressing climate change and sustainability. (Reflection notes from co-author Thomas Macintyre.)

Despite much fanfare around the signing of the peace deal between the Colombian government and the Revolutionary Armed Forces of Colombia (FARC) in 2017, deep socio-ecological challenges

are coalescing at many levels in Colombia [1]. Climate change affects the country through increasing droughts and floods [2] and increases social polarization around the politics of the peace deal alongside rising inequality, is disrupting social cohesion [3]. Although there is a lot of talk about the need for peace and a more sustainable world, the great unknown underlying this global challenge is how a more equitable and just future can be achieved.

As the UNESCO's 2016 Global Education Monitor Report [4] makes clear, education needs to play an important role in the necessary transformations towards more environmentally sustainable societies. Education shapes values and perspectives, and contributes to the acquisition of skills, concepts and tools that can be used to reduce or remove unsustainable practices [4]. However, with the complexity and ambiguity of sustainability proving difficult to address, there is increasing appreciation that we will need to learn to make our way toward sustainability [5]. Furthermore, it must be recognized that education in the context of sustainability and sustainable development can also become hijacked or neutered by forces in society that do not seek a fundamental change, but rather a way to keep things from changing to maintain power [6,7].

However, the Colombian educational system (especially in rural areas) is struggling in this endeavor, with low coverage, lack of quality and equity, and is searching for ways to make education relevant for rural communities and responsive to the social and environmental needs of its population [8]. Furthermore, at a more general and global level, there are concerns that much of education today is mainly preparing learners to function well in a globalizing economy that is based on continuous growth and expansion, thus unwillingly at best, accelerating unsustainability [9]. Hence, creating education that contributes to sustainable development raises both questions of quality (what constitutes education, learning and capacity-building that will lead to a more sustainable world?) and of transgression (how can stubborn systems and routines that normalize colonial and associated neoliberal practices, also in education, be disrupted?). Here we point at an apparent tension between the emancipatory intent of education, focusing on capacity-building and autonomously and critically finding ones' way in the world, on the one hand, and the instrumental intent of preserving the Earth for all people and species. While, in theory, transgressive and transformative learning as a process can be of an emancipatory nature, without having a specific outcome in mind, in practice there will always be some normative end which makes it instrumental as well—the goal of emancipation in itself is instrumental. The point then becomes to be explicit about the normative direction and the moral compass that guides the educational pursuit [10].

While lead author Thomas Macintyre sits with the students at the University of Quindío, co-researcher Tania Villarreal is in the region of Putumayo, sitting with an indigenous *Camentza* family around the *shinyak*—the ceremonial fire of the family. Tania in her profession as social innovator has been working with this Indigenous family, and their surrounding community, for three years in developing a shared vision for their community organization. Sitting around the fire they are all drawing a biocultural calendar for their community as a means to connect tourism to the cultural events and biodiversity of the region. Meanwhile, in the coffee region of Colombia, Martha Chaves is with the association of peasant farmers of Filandia (ANUC), discussing the creation of a tourist route between their farms offering visitors an authentic experience of the Colombian peasant culture. ANUC works toward promoting the interests of peasants, whose livelihoods are being threatened through the absence of effective government regulation, which is lowering the price for the goods they produce, while mass tourism to the town of Filandia is leading to gentrification of the surrounding countryside. Last, off the Atlantic coast, Margarita Zethelius is with the afro-Colombian community members on Islas del Rosario, meeting at a community center *La Casa Taranga*. After gaining status as an autonomous afro-Colombian community, the afro-community are working towards integrating their local customs and sustainability efforts such as mangrove restoration into a tourism offer.

These experiences and interactions all contribute to the making of a community-learning course called *Turismo de Origen* (Tourism of Origin). This course is a result of the three year international T-Learning project ("Transgressive Social Learning for Socio-Ecological Sustainability in Times of

Change," funded by the International Science Council. See www.transgressivelearning.org), focusing on transgressive forms of learning-based change at the nexus challenges of food security and sovereignty, water and energy, and social justice context of climate change [11,12]. In the context of this research, we are interested in what transgressive learning looks like in practice—especially the dilemmas of how to initiate and engage in participatory-community research without dominating the process as academics.

The Colombian case study of this international project has involved several of the co-authors working closely with grassroots initiatives in the fields of sustainability and climate change [13]. As will be described later in this paper, participatory diagnostics in grassroots communities resulted in responsible tourism being chosen as the umbrella theme for this course, with a shared challenge being the threat and opportunities of tourism to the initiative communities. Designed as a community-learning course involving different sectors of society and forms of learning [14], this paper will focus on the inter- and intra-learning taking place between the following two groups (see Figure 1 below):

1.   Academic participants and organizers: The representatives of higher education are the core Koru team (first five co-authors), and the bachelor and alumni students in the field of Social Work from the University of Quindío. The students were selected because of their active engagement in social justice in the region of Quindío, such as activism against mega-mining, which connected well with the course focus socially engaged and equitable form of tourism. The organizational teams (described in Figure 1, below) connect each of the participating community initiatives with a co-researcher acting as a facilitator.

2.   Grassroots participants situated in three different Colombian contexts (see Figure 1 below), all challenged by the detrimental effects of mass-tourism, which is affecting Colombia due to the opening up of the nation to tourism [15], and leading to the wider, global phenomenon of unequal power relations between "hosts" and "guests" due to violence and dispossession [16].

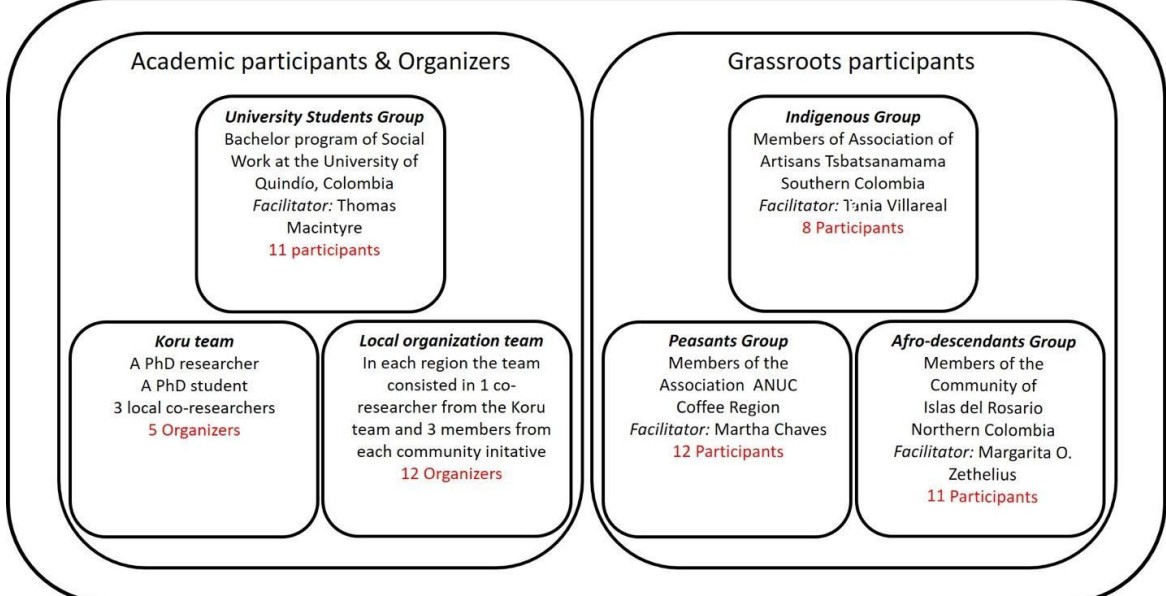

**Figure 1.** Characteristics of the organizers and participants in the course *Turismo de Origin*.

During the design of the course *Turismo de Origin*, an educational approach began to emerge, which was sufficiently interesting in its ability to bring different forms of learning together, to merit further development. We have called this educational approach *Koru*, which we present as an innovative approach for designing, implementing, and evaluating a community-learning course. Through presenting a practical application of the *Koru* approach, the overall objective of this paper is to explore the levers and barriers to bridging forms of learning across the diverse contexts of the course

participants, particularly that of higher education (represented by the *Koru* team and the University Students Group) and community-based learning (grassroots participants).

## 2. Conceptual Background

### 2.1. Transgressive Learning and Decolonizing Pedagogies

Transgressive learning connects well with currents of decolonizing pedagogies, which are gaining ground in academia as a resistance to Western imposed paradigms of development [17–22]. According to Lotz-Sisitka and colleagues [11], what is important in these theories of decolonization is the emerging transformative praxis in the sustainability sciences necessary to promote forms of learning that fundamentally address the challenges of climate change. It is important to note, however, that decolonization specifically refers to a historical process whereby countries that were colonized by foreign powers obtain their independence. Decolonizing pedagogies therefore refer to pedagogies that promote marginalized forms of knowledge, such as indigenous and local knowledge (ILK), which has a strong tradition in Latin America in line with Freirean emancipatory pedagogies and environmental education in Brazil [23–26], alongside African movements [21,22].

In the context of higher education and curriculum studies, decolonization tends to either refer to fundamental changes in the nature and identity of the university—questioning and dismantling oppressive colonial structures in institutions—or refer to the changing content of the curriculum to include indigenous and local knowledge, which is more relevant to regional and student context [22]. Overall, a perceived danger in decolonizing pedagogies is that emphasis is placed on replacing a dominant paradigm with a marginal paradigm, both of which are considered homogenous and static [22,27]. This fails to recognize that all knowledge traditions are saturated with power and inequalities, and that rather than being static, knowledge traditions are constantly changing through an exchange of ideas and practices. We can understand this dynamic as an ecology of knowledge [24] in the form of epistemological pluralism [28].

In response to these challenges, transgressive learning takes a dynamic perspective on knowledge, one that recognizes that dominant power structures and persistent inequalities act as barriers to the realization of more sustainable futures [9,11,29–31]. The contribution of transgressive learning to decolonial debates is that rather than replacing dominant with marginal paradigms, the focus is on the learning processes that can disrupt or transgress often highly resilient "locked-in" systems [32] and can thereby open up possibilities for transformation of the system itself. Such learning processes require a certain comfort with and even appreciation of the uncertain, changing and often unknowable aspects of knowledge (co)creation, in what we can understand as a learning ecology [33]

### 2.2. From Epistemicide toward a Learning Ecology

While Higher Learning Institutions (HEIs) have traditionally been regarded as the main source of knowledge production, and remain powerful actors necessary for socio-ecological transformations, they are increasingly being critiqued for their inability to generate knowledge and solutions capable of fundamentally addressing wicked societal challenges. Hall and Tandon [17], note that part of this inability stems from HEIs tendency to exclude many existing knowledge systems, including many indigenous ones in the world, and as such, promote only a fraction of the knowledge in the world. Referring to the work of Sousa Santos [24], Hall and Tandon go as far as to describe this process as epistemicide—the killing of other knowledge systems.

A response to this critique of epistemicide is adopting a more responsible and responsive ethos to fostering the renewal of higher education [34]. Based on the educational design principles of educating whole persons, as presented by Tassone and her colleagues [34], the head-heart-hands model of transformative learning [35,36], and the more transgressive form of "learning to anticipate," [37], the following forms of learning represent some elements of a learning ecology [38,39], which we will connect to the bridging objective of this paper between higher education and community learning.

1.  Learning to know—The head-based cognitive form of learning which we traditionally connect to classroom-based teaching. Cognitive learning is needed to research complex socio-ecological issues, to understand multiple forms of knowledge, to navigate uncertainties, and reflect on innovative solutions. Beyond knowledge transfer, this domain also explores deeper transformative learning brought about by critical reflection.
2.  Learning to be—This is the affective "heart-based" domain of learning related to emotions, feelings and relational knowledge [35]. It is brought about by collaboration between actors, cultivating social attitudes and values through a sharing of experiences. This is connected to "learning to care" [40], with a focus on establishing deeper connections with people, places and other species, and developing an ethics of compassion, empathy and care [41].
3.  Learning to do—The psychomotor domain of learning, or the physical expression of our capacities through our hands. This domain includes learning practical skills and physical work, with an important aspect being physically present, and building relationships with a place [35].
4.  Learning to anticipate—The domain of learning for an unknown future [41,42]. Focus is placed on being critical and reflexive to what is or is not being learnt, absences and unintended learning outcomes [37], as well as on finding ways to resist, uproot and transgress structural barriers to change and transformation [9,11].

## 3. Methodological Background

### 3.1. The Koru Approach

The *Koru* approach evolved out of an action-research methodology, which in broad terms involves scholarship–practice with a focus on multi-stakeholder engagement and a change agenda [43]. Specifically, *Koru* has been guided by earlier work on Transgressive Action Research (TAR) in the T-Learning project, which is characterized by a focus on working with co-researchers on local sustainability issues, employing participatory tools to promote transgressive learning through critical and reflexive thinking and action [13].

The use of TAR poses methodological challenges as the external researcher is (or becomes) closely connected to the internal researcher, and is often sympathetic to the goals they pursue. This close connection and compassion requires the external researcher to be very transparent in terms of his/her own biases, values and assumptions, and to "bracket" them [44] in order to not be blinded by them and to be able to look at the data from a certain distance. Guided by criteria for action research [45], actionability and reflexivity, in addition to being systematic and the sound application of methods and analytical tools, have been critical elements in ensuring a high quality in our research.

Part of the TAR project was carrying out a future search exercise in 2017 called Dragon Dreaming [46], whereby the first five authors reached a collective vision of generating pedagogical material in the respective communities they were working in. This material was to be organized into an educational course to offer other communities [47]. The focus of this material was community-based knowledge and experiences in climate change, with material being produced in transformation Labs (T-Labs) held in participating communities in 2017/2018 [13,48–55]. With T-Labs having been completed, the challenge was identifying mediums to connect these experiences between community initiatives, as well as to other actors such as academia, so as to promote post-colonial epistemologies and curriculum innovation that would unite action and reflection around local contexts. The following are the three learning concepts which came to characterize the Koru approach:

Blended learning—As a means to connect geographically diverse participants in the course, without losing the personal connection between participants, it was decided to work with connecting face-to-face learning with Information and Communication Technologies (ICT), known as blended learning [56]. A collaboration was developed with the Colombian company *Nuevos Medios*, who provided technical capacitation to the *Koru* team in how to develop online learning modules and manage registrations on the company's learning platform. Nuevos Medios is a Colombian company

that specializes in developing online learning platforms. The company provided support in the use of their online platform KME 360, including how to add and organize learning modules, inscribe participants, and carry out evaluations. (See the website http://www.nuevosmedios.net/ for more information.)

Peer-to-peer learning—Interactive learning between participants was based on the Farmer to Farmer learning (FAF), developed by the agroecology movement in Latin America [57]. This involved (1) clearly identifying the main interests and necessities in terms of what community members wanted to learn through participatory diagnostics; and (2) the generation of pedagogical material within community contexts, centered on community experiences, needs and what they have to offer other communities. Through this learning concept, the model of expert knowledge and subject is blurred by giving voice to knowledge and experiences at the grassroots level, alongside expert knowledge.

Project-based learning—Experience from the T-Labs demonstrated the importance of basing learning around practical projects relevant to community initiatives. Project-based learning was therefore used, which involves inquiry-based learning through practical projects that reflect participant knowledge and experiences [58].

Inspired by a diagram of project-based learning [59], the *Koru* approach merges the learning concepts above into a 10-step process to design, implement, and evaluate a community-learning educational course (see Figure 2 below). Each step is led by the core *Koru* group, which is composed of co-researchers acting as cultural facilitators between each community group and the overall project. The core *Koru* group is in charge of the organization and facilitation of the course with the communities' input and consent. In addition, some steps involve local teams, which are composed of community members and their respective co-researcher.

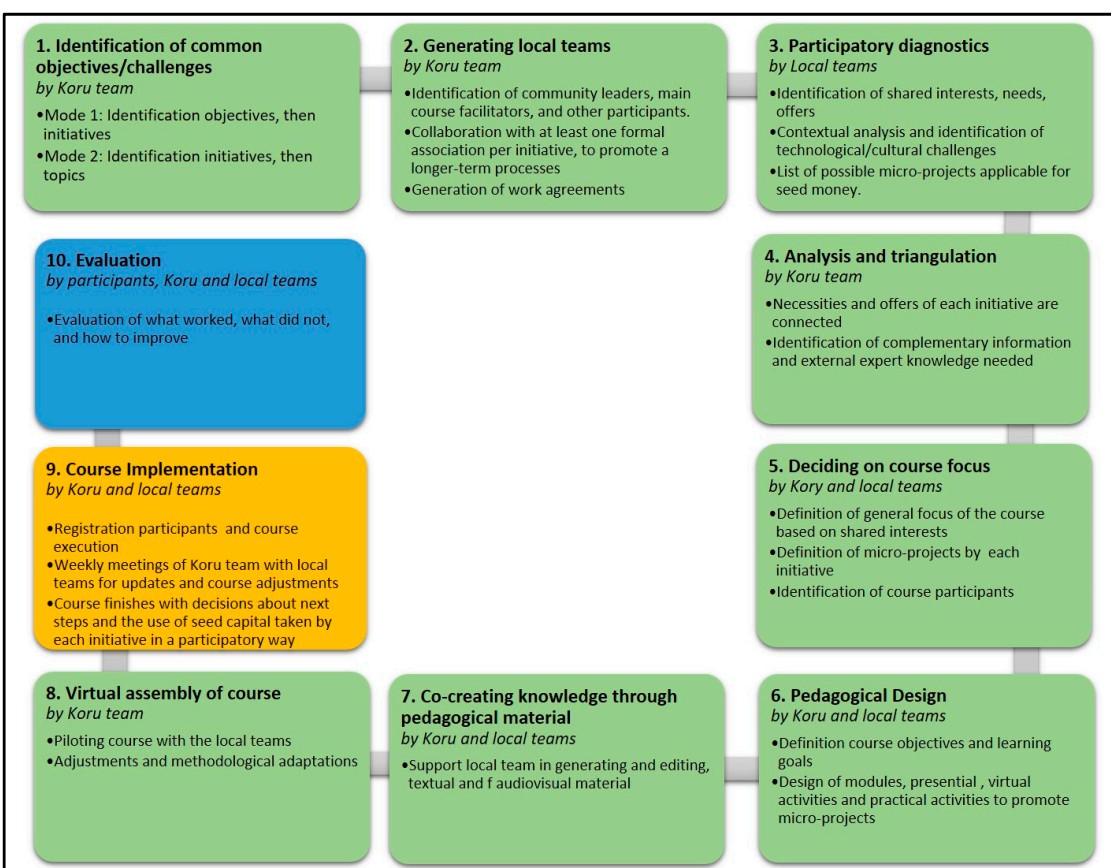

**Figure 2.** The 10 steps of the Koru educational approach. The green steps (1–8) represent the design stage; yellow (step 9) represents the implementation stage; and the evaluation stage (step 10) is represented by blue.

*3.2. Methods and Data Analysis*

As a means to explore the application of the *Koru* approach in transgressing boundaries between community learning and higher education, we have divided the methods section along the three stages of design, implementation, and evaluation. We will first present the *design* stage (steps 1–8) as a descriptive account of how the learning concepts were operationalized in the development of the course *Turismo de Origen*. These descriptions are based on a review of collaborative documents developed by the Koru team throughout the course process. We will then present the methods used for analyzing the implementation and evaluation stages, whose results are presented in the results section.

### 3.2.1. Step 1: Identification of Common Objectives and Challenges

In the third year of an action-research project, the co-authors of this paper have been working closely with the grassroots initiatives in the present research in the fields of sustainability and climate change in Colombia [13]. Through transformation workshops held with these initiatives, and in some cases long-term relations between co-researcher and the initiative, trust and engagement had developed as an academic–non-academic partnership. The *Koru* team decided to follow mode 2 in step 1 in Figure 2 above, whereby the initiatives were chosen first, and the course topic was chosen afterwards following a more participatory approach to defining the field of inquiry.

### 3.2.2. Step 2: Generating Local Teams

An important aspect of the *Koru* approach is working with local co-researchers who both understand the day-to-day challenges faced, while also understanding the context and requirements of the research project and communicating this to community members. With the project respecting the knowledge and experiences of co-researchers, the latter identified community leaders and other motivated participants who had the motivation to actively participate in the course, creating local teams. Concrete agreements concerning consent to audiovisual material was negotiated and signed, as well as the formal associations connected to each co-researcher insuring responsibility for the continued process after the course finished.

### 3.2.3. Step 3: Participatory Diagnostics

In line with the FAF methodology, participatory diagnostics were carried out by the *Koru* team in each community initiative to identify needs and offers. Recognizing different cultural contexts and values, a tailor-made process was carried out in each community initiative. In the *Camentza* community, for example, with their strong values on nature and medicinal plants, there were talks while walking through the productive plots of members (*jajanes*), followed by participatory mapping centered around territorial challenges. With the Afro-Colombian group and their values of solidarity, there was conversations with participants in *la casa Teranga* on how to network different actors in the region around territorial development, as well as mapping exercises around necessities and offerings at different scales. Finally, with the association of peasant farmers (ANUC), with their value on farm production, individual farm-resilience diagnoses were carried out, followed by a socialization of results, and a SWOT analysis and discussion of peasant farm resilience in the region. Complementing context-specific methods, Global Ecovillage Network (GEN) design cards, were used in the three communities, which explored holistic sustainability and whole-systems design using visual representations and short descriptions for sustainability (see Figure 3, below). (See the website https://ecovillage.org/resources/market/ecovillage-design-cards/ for more information on the Ecovillage playing cards.)

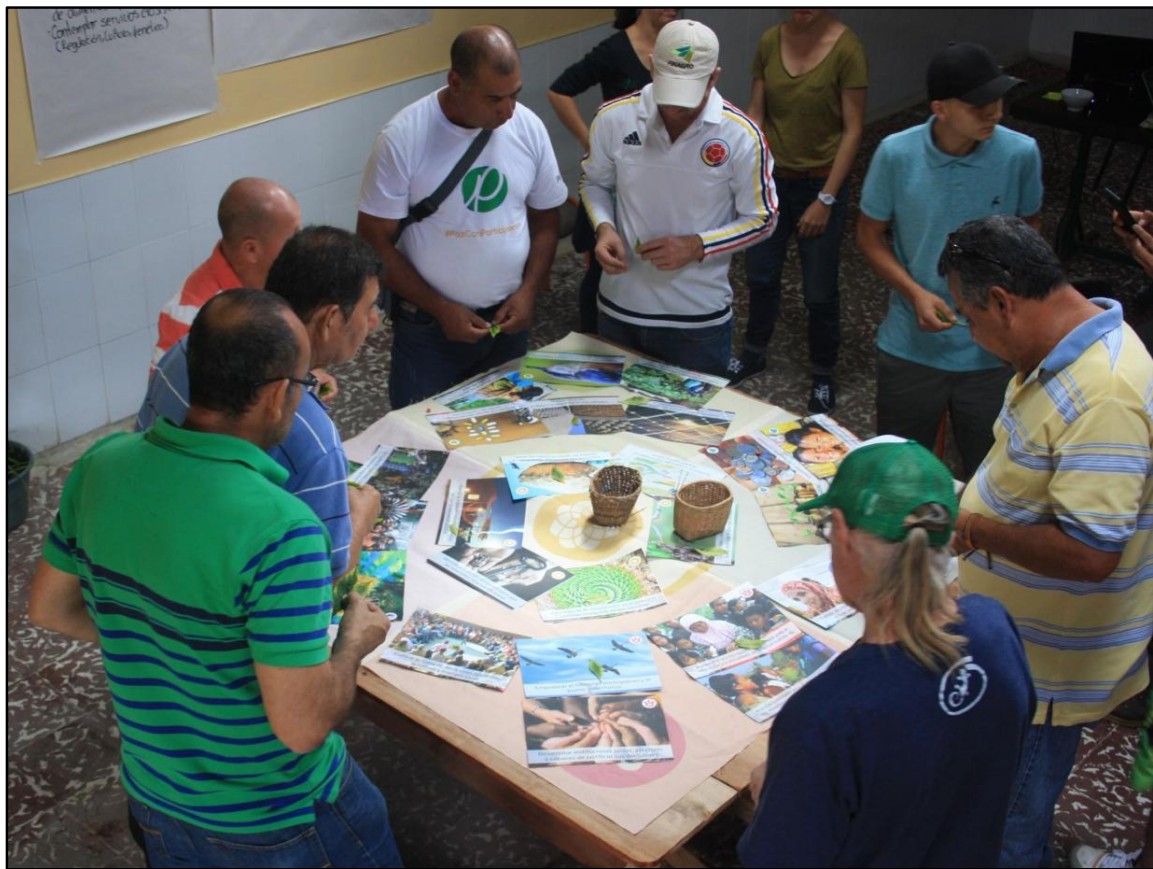

**Figure 3.** Global Ecovillage Network (GEN) card game used to diagnose strengths and weaknesses across different dimensions of sustainability. Photo credits: Thomas Macintyre.

3.2.4. Step 4: Analysis and Triangulation

Data from the diagnostics was coded and mapped by the core *Koru* group through connecting the needs and offerings of the three communities. Figure 4 below represents this mapping exercise whereby the blue circle represents the needs expressed by the Afro-Colombian community in Las Islas del Rosario, the orange boxes represents what the indigenous participants Camentza have to offer, and in the green boxes illustrates the offers expressed by the peasant participants of ANUC. The arrows represent the connection of needs and offerings between communities. Within the blue circle, the needs that are in red were those that could not be matched with other local knowledge, thus being paired by outside expert knowledge.

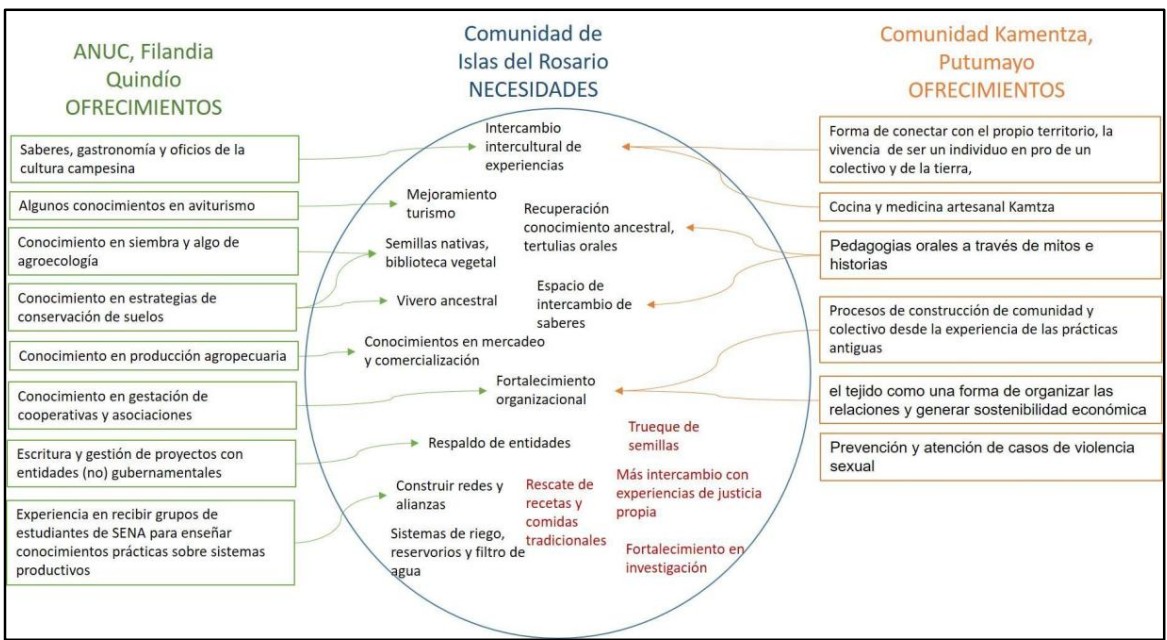

**Figure 4.** A map of the connections between the needs and offers of the three participating communities. It is not necessary to understand the Spanish, just how the need and offers were connected between the communities.

### 3.2.5. Step 5: Deciding on the Course Focus

Despite the diverse contexts of the three communities, a list of common themes emerged. The main theme was that of responsible tourism, with all communities sharing their struggles with reconciling tourism as a necessary source of income, versus its shadow side of cultural appropriation and environmental degradation. A shared desire was to use tourism as a means to promote local culture, increase biodiversity, as well as generate dignified livelihoods. *Turismo de Origin*—a form of tourism that promotes the roots and authenticity of the region—was suggested by an ANUC participant as a name for this form of tourism and was taken up by the *Koru* team and used as the course focus. Under the supervision of the lead author, a group of university students from the university of Quindío, Colombia, were invited to contribute academic skills of knowledge systematization between the participating groups, as well as representing academic actors in the dialogue with community members.

### 3.2.6. Step 6: Pedagogical Design

A five-day workshop was held with the *Koru* group in which the course objectives and learning goals were identified. This workshop also involved the design of the curriculum modules, including content and activities (see Figure 5, below). Each co-researcher took responsibility for developing one of the course modules, as well as taking responsibility for developing the community project each initiative would work toward, in line with the learning concept of project-based learning.

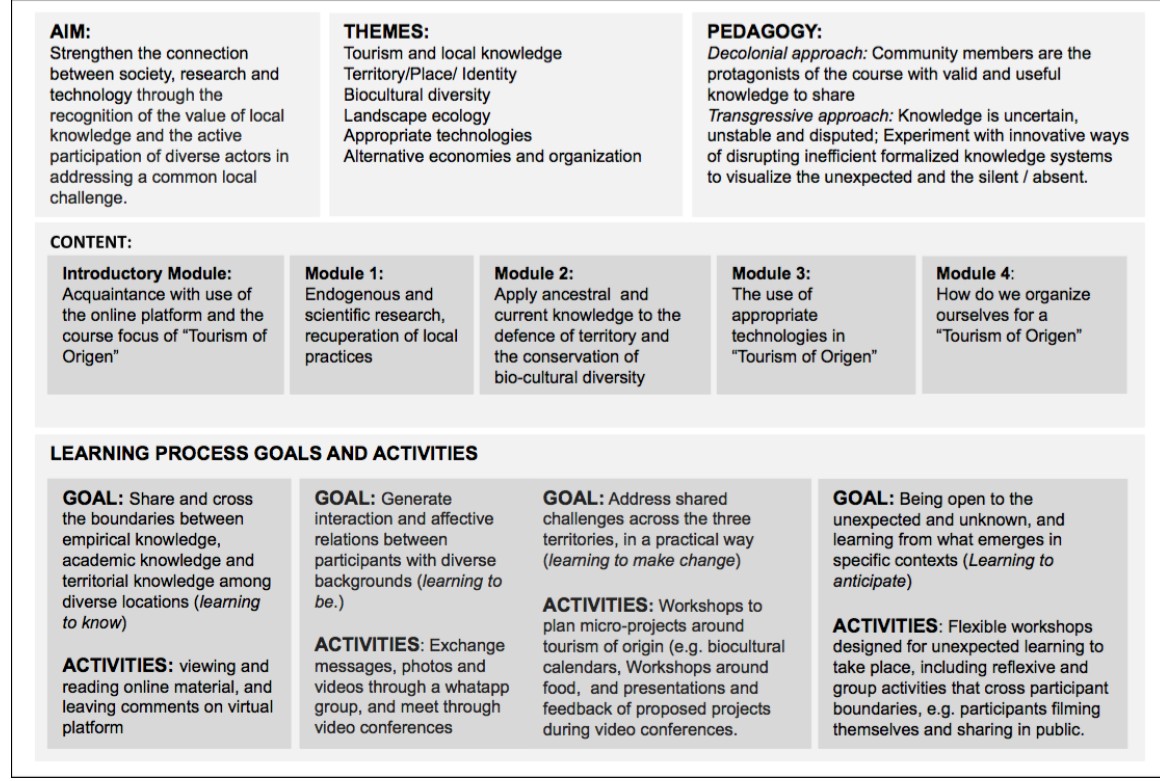

**Figure 5.** Overview of the course Turismo de Origin.

### 3.2.7. Step 7: Co-Producing Knowledge through Pedagogical Material

Following the peer-to-peer learning concept, the majority of pedagogical material was generated in the community initiatives themselves by local teams. Each local team used a video camera to film interviews with community members with respect to content modules, as well as more technical videos, such as water-catchment systems in Las Islas del Rosario.

### 3.2.8. Step 8: Virtual Assembly of Course

Each co-researcher assembled the module they were responsible for, including editing and uploading videos, adding text, and contributing to a virtual library where further resources connected to the course themes could be accessed by participants. The finished course was then piloted by each local team, with technical and methodological adjustments carried out to better align the course with local realities. An example of such an alignment was the shifting of responsibility of uploading material to the online platform from the local participants, who often had low Internet connectivity, to the co-researchers who generally had better internet connectivity.

### 3.3. Course Implementation

The implementation of the course (step 9) was analyzed through co-researcher narratives connected to the different forms of learning presented in Section 2.2. Narratives have been shown to be a useful way of communicating community learning [30], with narrative-inquiry research being especially useful in curriculum development [60,61]. This method also aligns closely with the Transgressive Action Research methodology, which seeks to bring in the voice of co-researchers [13]. Co-researchers wrote field notes during the implementation phase of the course, recording key learning points related to a specific form of learning.

*3.4. Course Evaluation*

The evaluation stage (step 10) is presented through the learning experiences perceived by course participants. Focus groups were conducted by the co-researchers with each of the three grassroots participant groups, and the university group (see Figure 1, above). Focus groups have been shown to be an effective way of eliciting the student voice in higher learning curriculum design [62]. Each focus group was facilitated by the corresponding co-researcher, with questions centered around the different aspects of the course and forms of learning (as described in Section 2.2 above) (see Appendix A for focus group questions).

The process of data analysis for the focus groups was based on thematic coding [63]. Each focus group was recorded, and an abridged transcript for each focus group was made by the lead author in the qualitative analysis software NVIVO. In order to enhance reliability [63], the coding process was performed in various rounds and by various authors. The lead-author carried out the first round of coding deductively, based on the focus-group questions template, with the objective of categorizing the data and related quotes of the participants according to the four forms of learning. Through nested coding, these categories were then coded inductively as to whether they acted as levers and/or barriers to bridging community learning and higher education. The second author revised the first and second round of coding, with some divergences in results being resolved through discussion among the two coders. In a third round, the previous categories were analyzed by the first two authors individually to identify the themes that were emerging based on the lever and barriers previously identified. There were some differences in the themes identified between the two authors, and these were discussed in a fourth round between the two coders, as well as co-authors six and seven, who did not participate in the course, but who contributed to the thinking process throughout the development of the research. As a result of this process, the final set of themes have been defined, and elaborated in the results Section 4.2.

## 4. Results and Analysis

*4.1. Implementation (Step 9) Narratives from Co-Researchers*

The course *Turismo de Origen* took place between 5 and 30 November, with 56 registered participants across the participant groups (presented in Figure 1). A challenging aspect of the course was the participation of individuals and communities with different social, economic, and cultural backgrounds, as well as technological access and capacity. As the following accounts demonstrate, these anticipated challenges led to a focus on flexibility and collaboration between co-researchers and participants, leading to intended and unintended learning experiences.

4.1.1. Learning to Know: "Learn Together, Not Apart," Margarita Zethelius, Las Islas del Rosario

"From the start I had anticipated challenges in terms of internet and electricity availability, as well as access to cellphones and computers to take the course. Las Islas del Rosario is not connected to the energy grid, and many residents have few economic resources. The logic of the online platform is that each participant registers with their email address, individually view material from their account, with the platform then providing individual statistical data for evaluation purposes. However, the phones of two of the participants in my group broke right before the course began, and during the inscription I realized that many participants did not have email addresses with which to register. It also just seemed difficult for many of the participants to carry out activities by themselves in the format of the modules. Unlike the participants in my group, I have a university degree, and have been taught to work individually and systematically. Throughout the course I noticed that the "traditional" logic of the participants is to *learn together, not apart*, and that for them it is challenging to navigate the more individualistic logic of higher education. I decided to adapt how the course was given by collectively showing the online material at defined times through a projector in *La*

*Casa Teranga*. This was much more motivating for the participants, as they could watch and discuss the material together. As one of the participants shared '*it is nice to hear what others understand and share opinions.*'"

(Figure 6)

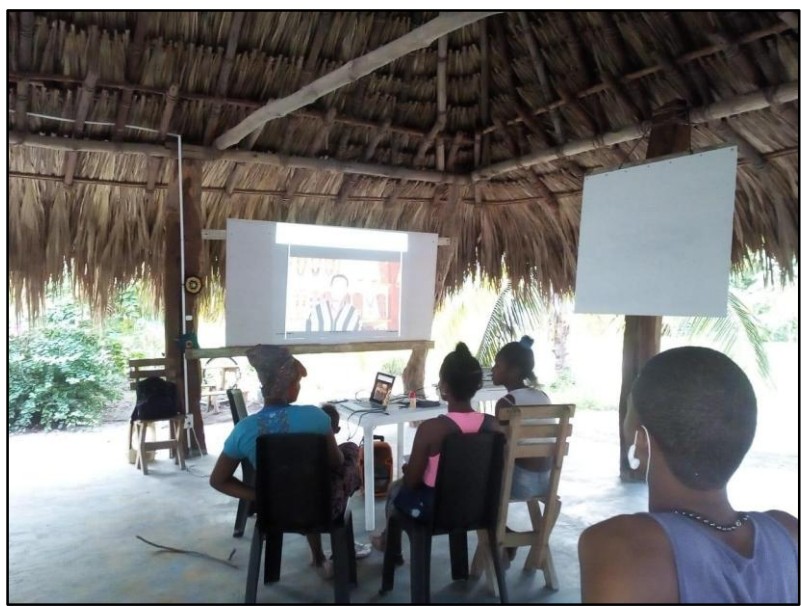

**Figure 6.** Participants from Las Islas del Rosario viewing material in *Casa Teranga*. (Photo credit: Margarita Zethelius.)

### 4.1.2. Learning to Be: "ICT Created Affective Bonds of Friendship Between Participants," Tatiana Monroy, Facilitator

"It is midnight and I am sitting by the fire at my home in Spain, ready to facilitate a video conference between the participants of our course. In different parts of Colombia, six hours behind, people are finishing work and will be getting ready for the conference. For the past month the course participants have only seen each other in videos on the online platform, and shared messages on WhatsApp. Then the time has come to start. I introduce everyone, explain the agenda and etiquette for using the video conference platform. Each community has elected a spokesperson to present their community and the project they are working on. I am very impressed by the engagement of the participants - the attention they are giving each other. Especially with the *Camentza* family I felt the depth and conviction of their real world situation, which is often not conveyed in classroom learning. Although this meeting has been virtual, I feel that the *ICT created affective bonds of friendship* between participants who would not normally meet."

### 4.1.3. Learning to Do: "Experiential Activities Can Lead to Collective Understandings," Martha Chaves, ANUC Filandia

"Experiential learning was best felt during the main workshop of the course, where my group, along with the students from the University of Quindío, met at the ANUC *Casa Campesina* in the town of Filandia. Each participant was encouraged to bring a food dish based on a recipe from their childhood, representing a traditional food of the region. Oscar presented a pork recipe that his mom used to prepare each Sunday, and Hernan brought a plantain *colada*, a beverage that filled him with energy before going to school every day. Discussions centered around how relations and emotions were built and expressed through food. This was exciting because the experiences crossed all boundaries between the peasants

and the students present, providing a feeling of a *collective understandings*. As a group we talked about the other communities in the course who had shared traditional recipes through the WhatsApp group and whom had emphasized the effort they put into keeping their culture alive. In relation to these reflections, an interesting sentiment shared by both university students and ANUC members was the need to improve the organization of ANUC to maintain the peasant culture in the face of mass tourism."

(Figure 7)

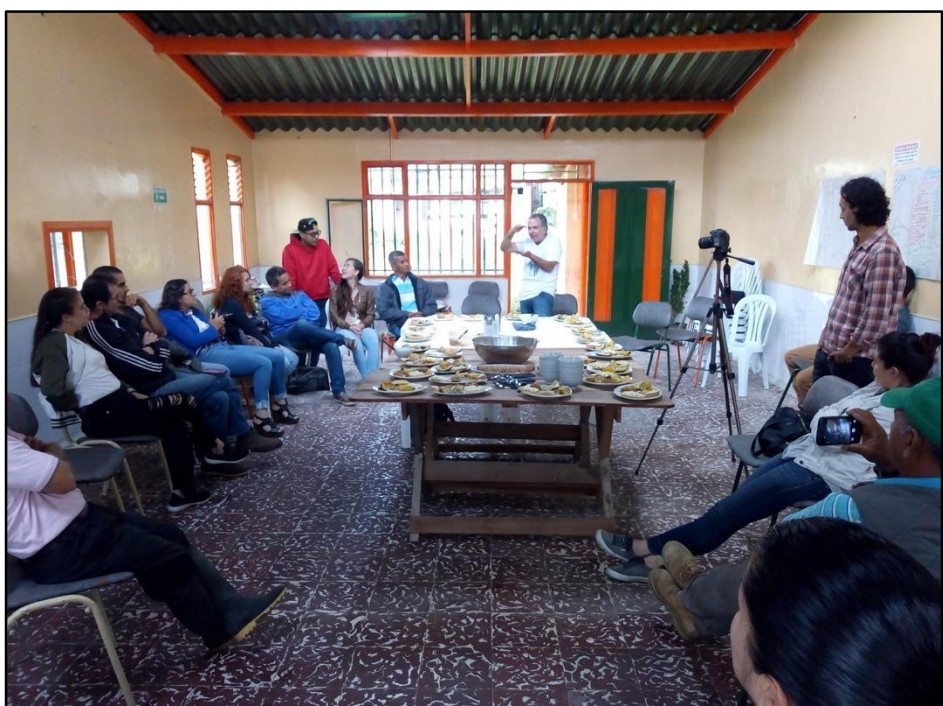

**Figure 7.** Members of the Asociacion Nacional De Usuarios Campesinos (**ANUC**) and university students of Quindío listening to local historian in *la Casa Campesina*, Filandia. (Photo Credit: Martha Chaves.)

### 4.1.4. Learning to Anticipate: "How We Value Our Own Knowledge in Relation to That of Others." Thomas Macintyre, Student Group

"I had expected the university students to be the most active members in the course, contributing and synthesizing concepts, as this is one of the great strengths of academia. Although the majority of students were active in sharing research experiences in the WhatsApp messaging group, and systematizing the projects of each community into a national tourist route, there was particularly one student who struggled to provide a voice. Having noted that the student was not participating in the WhatsApp group, I asked the student why not. The student replied, *"Who am I to comment on 'other' forms of knowledge? Other participants come from communities where there is so much culture. I have nothing to contribute with."* I suggested the student (who had completed all the online material) share views on scientific knowledge in academia. The student replied, however, that it was the community members who were the protagonists, and that the emphasis should be on what they had to share. As an academic myself, fascinated by other ways of knowing, and having shared my own perspectives with course participants, I saw this as an unexpected outcome of the course, demonstrating a disbalance between *how we value our own knowledge in relation to that of others*."

*4.2. Evaluation of the Course Turismo de Origen*

The evaluation of the course, as perceived by the participants, was examined through four focus groups, each facilitated by the respective co-researcher. The focus groups explored the learning experiences of participants engaged in the course, especially in terms of perceived levers and barriers supporting or hampering multiple forms of learning (see Appendix A for focus group questions). The following subsections illustrate the results of the thematic coding (method described in Section 3.1.) with each theme referring to the four forms of learning described in Section 2.2. (learning to know, be, do, and anticipate).

4.2.1. Use of Information and Communication Technologies (ICT)

The ICT theme emerged strongly in the course due to the blended learning approach, whereby technologies such as videoconferencing, WhatsApp messaging, and the online platform were combined with physical face-to-face workshops to provide opportunities for participants from different communities and students to interact across Colombia and Spain.

Various course participants shared the ICT mediated learning environment as an opportunity for learning and for capacity building (learning to do). For example, Emerenciana from the Camentza community noted in the focus group, "I was intimidated by the technology for the course, but you [co-researcher Tania Villarreal] came to our house and showed us how to use the technology, it was a very good opportunity." In other cases, ICT was even considered as a means for valorizing culture and ancient knowledge (learning to know). For example, Marcella from Las Islas de Rosario noted how as a child, "when teachers asked us for information, they sent us to the library or to ask somebody. Now we ask everything to the computer, the smart phone. However, during the tertulias [oral gatherings], we recorded the conversations of the elders, we are giving a positive use to the technology. Giving voice to the *saberidores* [those that know]."

Furthermore, several participants experienced the online learning platform and material in a way which we can understand as supporting critical reflection (learning to know). For example, Camila of the *Camentza* community noted of Las Islas del Rosario, "From a situation of so much scarcity, they continue moving forward, not braking, although they do not have energy, or their own drinking water. Here we have everything, and we do not appreciate it. I saw on the videos on the platform they have ways of harvesting water, while here we complain if the water gets cut off for two hours in the evening." The environmentalist Fernando of ANUC, in his reflections about the afro community Las Islas del Rosario, said, "we sometimes think of the *costeños* [people from the coast] as very 'tropical' [relaxed work ethic], but in the videos I saw people with a lot of energy and drive, conscientious and worried about their future."

On the other hand, some participants reported that the learning material provided too much information (learning to know). For example, during the focus group, social work alumni Alejandra from the University of Quindío shared the following: "it was fun that most material was in the forms of videos, but I got saturated by the quantity of visual material." This feeling was collaborated by others, for example, by co-researcher Margarita during a feedback meeting with co-authors, whereby Margarita noted that participants in Las Islas del Rosario started losing focus after watching a series of videos, especially the videos which only showed the person speaking.

Alongside the above experiences, some participants using online technologies for the first time, shared that they unexpectedly realized that technology is changing the world, and can help empower local communities (learning to anticipate). For example, the elder peasant Guillermo from ANUC shared his amazement that the facilitator for the video conferences was in Spain: "She [co-researcher Tatiana Monroy] could connect us to the community in *Las Islas* and the *Camentza*. This is something amazing. How the world is changing." Or for example, Emerenciana from the Camentza community noted: "We talk a lot about what is one's own, what is sacred and every-day. But this also needs to move in the world of technology ... bringing these aspects together will help us give some authority in the work we are doing, especially with the institutions we are working with."

### 4.2.2. Relating to Place and One Another.

Relating to place and to one another was another relevant theme emerging from this community-learning course supporting learning in situ. Several participants expressed an awakened sense of connectivity towards their territory and each other (learning to be). For example, Guillermo from ANUC noted, "We are so influenced by technology, this course grounds us in reality. Sometimes we do not valorize what we have; our love for our territory, the ecosystem." Similarly, Yudy from the community Camentza said, "Today, society is moving so fast, things are so ephemeral, we are convinced of models so contradictory to what is our own, this system of consumption, consumption and to have more, this thought makes you forget who you are as a person, the most basic, your family . . . it is necessary to return to knowing which territorial space you are inhabiting, where you are living." This affection for place was also highlighted by student Angelica in reference to the youth on Las Islas del Rosario: "I saw in Las Islas that the youth were the eco guides, who were focusing on the coral reefs, that the youth are appropriating their territory, loving their territory, they want to stay and not leave for big cities where there is less of a social fabric."

Overall, a focus on place and relations generated a strong self-reflective environment (learning to know), according to participants. For example, alumni Alejandra stated, "I was very impressed by the associativity in the indigenous Camentza community. There was a moment when one of the older women was explaining how the family all gather around the *shinyak* to talk and discuss important topics. Getting to know their discourse and their interpretations, and the symbolic part is very beautiful–I liked it a lot because I have not heard such things before, I did not have this knowledge, and was left with the impression that they have certain things very well clarified, while ourselves, and the campesinos here [in Filandia ANUC] are lacking these clarities." Guillermo from ANUC noted something similar through a reflection on the relationships in the Camentza community: "The terms they use are always in the plural, "We are. We do. We make" . . . Here in Filandia there is a mentality of individualism, typical of the coffee region culture: the producer has their own coffee, own pulping machines. "I have this, I do this.""

This self-reflective environment created by the course also generated ideas for new ways of planning and doing things (learning to do). For example, Yudy from the Camentza family noted that the course " . . . has created a very strong reflection, because often we are working on certain topics, but we do not relate these topics to what is happening outside of our territory—as they relate to other communities. These are reflections which we should take advantage of, knowledge and experiences that they have. What more could we do in our own territory? We have a plan to reach a vision we have, but what about the experiences of other communities. It gives a much bigger plan for what we can achieve."

### 4.2.3. Intercultural Communication and Interactions

The importance of intercultural communication and interactions realized through bringing together different cultures and voices, came out strongly throughout the course. In the focus groups, several participants expressed the relevance of understanding and learning from eachother's cultures (learning to know). For example, alumni Alejandra shared that she enjoyed the course methodology of having three different communities each with "their own cosmologies, ways of understanding the world. The course helped us understand each one of them." Likewise, "Marcella" from Las Islas del Rosario stated "I found it interesting to learn about other geographic spaces which I did not even imagine existed, and communities like the Camentza." In his reflection, Juan Juagibioy, a youth in the Camentza community noted how the course "made me think about all the other communities in Colombia that exist. The course helped change my perspective that it is not only Sibondoy, ourselves as indigenous people, but a great diversity of cultures, ethnicities, ways of learning."

In other cases, participants reported that intercultural encounters helped to appreciate the deep commonalities among diverse communities (learning to be). For example, Yudy from the community Camentza shared that, "Every part of the world has different people, different characteristics, different

way of thinking. Look at the peasants in Quindío, and the people in Las Islas, bringing them together they have something in common with us, we all complement each other, we are living on one earth, with one Mother, and if you were put in this place then it is because you need to respond to this place, protection, to serve Mother earth, to care for her."

While intercultural exchanges facilitated learning between communities and for the student group, there was little mention from grassroots participants about student contributions, and few indications that communities were learning from the students (learning to anticipate—in terms of lack of learning). For example, questioned about the student group involvement, Fernando of ANUC noted the student participation was important, but it "could have been greater . . . perhaps because of their lack of experience." Guillermo from ANUC followed up on this remark with the following perspective: "Many times the expert comes to the farmer and the farmer thinks that the expert knows more because they have been to the university. We are ashamed to say anything because we think we do not know. But sometimes the academic comes and lowers themselves so much that there no real exchange. The challenge is to speak at the same level." This disjunction in learning as equal partners was shared by alumni Alejandra: "More than from the point of being an academic sharing our knowledge to the communities, I felt we were learning more than we were sharing. One has to look at what one can contribute with, but one knows that one has a long path to go. The communities have been down a long path, and much more than just theory." Student Angelica put it succinctly: "The protagonist of this course was the communities themselves, compared to other courses where the protagonists come from outside. Outsiders come and present their knowledge. They are the ones telling the story, teaching the communities."

A reason for this was provided by alumni Alejandra who noted that more could have been done to engage participants in collaborative actions and dialogic interactions (learning to do): "More time was needed for the course, and there could have been more gatherings, more themes to discuss between us, this was part of the reason we did not manage a good dialogue with the communities . . . so we learnt from them, got to know them, but did not manage to achieve the collaboration and interaction desired. The course should generate a strategy that the students, like the communities, could go through and reflect on the material together, as it ended up being an individual reflection [instead of a collective reflection]."

## 5. Discussion

In the search for a more sustainable world, a key challenge is to find learning pathways and environments that can help overcome systemic global dysfunction by cultivating more relational and transformative ways of being [11]. This is also a challenge for higher education as it seeks ways to integrate sustainability issues, mindsets and capacities into its teaching and learning fabric. There is an increased recognition that this cannot be done in piecemeal, add-on and ad-hoc ways—for example, through sustainability courses—but rather calls for approaches that are both more radical and more holistic [64]. Latin American decolonization discourse seems to be highly congruent with the radical critical element of this re-orientation [65], while discourses on learning-based transformations and more relational forms of collective learning seem to be congruent with the holistic element [66].

Important for learning-based transformations is the notion of boundaries, whereby it has been argued that it is at the boundaries that learning takes place [67]. Transdisciplinary research attempts to transcend "disciplinary" boundaries [68] through, for example, academic and non-academic research, such as in the present study. Likewise, based on the transgressive action research (TAR) methodology, the *Koru* approach has been presented as an innovative approach to blend diverse learning concepts into a participatory, community-learning approach.

However, there are many ways of understanding the decolonization of curriculum, with calls for deeper reflective thinking about the content of courses, who takes them, how they are taught, and how we learn together [22]. Fundamental to this are issues of knowledge democracy and epistemological justice, whereby community-based participatory research has been put forward as a means to bridge

different ways of knowing, and therefore different ways of learning, to challenge underlying power relations between actors in higher education [17].

The course focus of *Turismo de Origen* was to ground a decolonial discourse in the local contexts of the three participating communities, aiming to valorize and bridge traditional, ancestral and academic forms of learning in the development of practical projects, which support participant needs and requirements. This involved engaging in dialogue with academia, and promoting transformative interactions between people and places as important principles for generating collective learning experiences [66].

In the following sections we discuss the themes which emerged in the results section above, describing them as barriers and/or levers to bridging higher education and community learning.

### 5.1. Use of Information and Communication Technologies (ICT)

Technology-facilitated learning is increasingly common in higher education, with interest in how it can catalyze reflection and deep learning [69]. Results show that in the *Turismo de Origin* course, ICT was a strong lever to bridging forms of learning between participants. In terms of learning to do, the ICT mediated environment provided an opportunity to communities to build their capacities and develop digital skills. From a learning to know perspective, ICT played an important role in facilitating the sharing of information between the different participants through the mediums of the virtual platform, the messaging service WhatsApp, and the video conferences. This acted as a learning bridge between academic and non-academic participants in promoting the competence of critical reflection, which is an important part of higher education in addressing socio-ecological challenges [70]. Technology contributed to critical thinking and self-awareness through promoting social interaction amongst participants across geographic and cultural strata, and even became a means for giving voice to elders. As explained by Barak [71], an important aspect of generating such reflection is the possibility of transforming the traditional role of the instructor as the main source of information and power, towards being a facilitator of the learning process. Technology acted as a bridge to generating anticipatory learning in the course, opening up learning possibilities not before encountered.

On the other hand, results indicate that the high amount of visual information provided through the ICT environment created a barrier for some, in terms of learning to know. Participants at times felt saturated by the amount of information provided. Although not manifested in participant reflections, the co-authors of this paper have reflected further on the risk that new technologies pose. On the one hand, the *Koru* approach is carefully designed to pay special attention to informed consent amongst participants, which is a sensitive topic in indigenous communities [72]. As has been demonstrated, indigenous communities can positively adopt and make use of new digital technologies according to traditional knowledge [73], which we believe occurred during the course *Turismo de Origin*, as testified by participants. On the other hand, it is important to note the detrimental role technologies can play in learning environments. Technologies such as smartphones can be highly addictive and form a distraction in educational environments [74], creating a distance between people and between people and place, that in the worst case plants the seed of the kind of system and culture that created the technologies, which in the end might jeopardize the diversity sustainability is calling for [75]. The use of ICT needs to be critically assessed and continuously monitored in light of these risks.

### 5.2. Relating to Place and One Another

Connecting project-based learning and peer-to-peer learning was an important means of recognizing concrete experiences of communities grounded in shared histories, stories, and local projects within a politics of place [76]. This provides possibilities for experiential learning activities, thus fostering sustainability competencies in higher education through preparing students for the real world [77]. This is especially relevant when combined with a critical pedagogical approach [76,78], which, according to Paulo Freire, seeks to decolonize through "learning to perceive social, political, and economic contradictions, and to take action against the oppressive elements of reality" [25].

A focus on place and relations acted as a lever to bridging forms of learning between communities as well as with the university students. In terms of learning to be, such focus has awakened participant connectivity to the place they inhabit, to each other and to who they are as people. They started valorizing what they have, the place they live in, the ecosystems and the other people. At a learning to know level, a strong reflective environment was generated. Participants reflected on their own connection to their territory through learning about other participants and their situations. This encouraged them to reflect on their own assumptions and expand their knowledge, also in connection to their own place, opening up new possibilities towards community action. Learning in situ, from each other, and through projects stimulated also desire for learning to do and making new plans in their territory, especially through building on the experiences of other communities.

Experiential learning activities with a focus on territorial projects acted as a lever to learning across the participants as they became motivated and engaged in the course, and could use what they had learned to do something that has an impact on others, especially their local community [79].

*5.3. Intercultural Communication and Interactions*

With the realization that socio-ecological challenges are complex and messy, there is increasing interest in transdisciplinary research approaches which connect academic and non-academic forms of knowledge in a problem solving and reflexive way [68,80]. Such approaches aim for the co-production of knowledge, often based on situated knowledge [81]. However, such approaches often face structural barriers in their implementation, such as institutional, organizational and cognitive differences between participants and organizations [81]. This can take the form of the marginalization of indigenous knowledge systems [17], as well as unequal power relations between academic and indigenous and local knowledge [82]. There are therefore calls to decolonize the transdisciplinary research approaches [21].

The course *Turismo de Origen* provided an illustrative example of how intercultural encounters have the possibilities to both act as levers and barriers to learning. At a cognitive learning to know level, the exchange between grassroots participants and students opened up a strong appreciation for new perspectives on the diversity of cultures in Colombia, as well as critical reflections on cultural practices. At a learning to be level, affective relations were generated between the grassroots community members, whereby their community experiences were inspirational for one another. If we think of heart-based relational knowing as the awareness of the relationships shared with community and the natural world [35], then there was strong relations built between the three communities.

From an anticipatory learning standpoint, there appears, however, to be underlying power relations and inequities between the participants that acted as barriers to learning. As made clear by the student group, the communities were the protagonists—they were the ones telling the story, instead of that role usually being assigned to the expert. This emancipatory approach to community learning perhaps contributed to what one student stated in the narrative section (Section 4.1.4) "Who am I to comment on 'other' forms of knowledge?" This suggests a feeling of marginality, a feeling that what one knows has less worth than someone else's, and does not deserve to be voiced.

It could therefore be argued that to some extent, the dominant form of western knowledge (represented by students and the *Koru* team) was exchanged for what was perceived as a more important form of community-based paradigm. Although this demonstrates the high degree to which the community participants felt empowered to express their own ways of learning, in what was initiated by "expert" academics, this in effect meant swapping a dominant "expert" paradigm with a marginal "community" paradigm, which is a risk in decolonizing pedagogies [22,27]. This, to some extent, negated the objective of transgressing the forms of learning between the different participants. On the one hand, this suggests the need for improving the participatory design of the curriculum, focusing more on participation by the students in the design of the course, and including more innovative activities to connect students amongst themselves and to community participants. On the other hand, the lack of connection between students and community participants may also be due to the perceived fragile and marginalized nature of communities and their forms of learning. Although

Le Grange [22] notes the importance of transdisciplinary knowledge in decolonizing the curriculum, addressing the replacement of one paradigm for another, there are clearly embodied power differences between participants of different backgrounds which make this difficult.

### 5.4. Limitations in this Research

During a joint retrospective reflection between the co-authors of this paper, it was agreed that such inherent power differences should have been addressed from the start of the course through a delegate assessment of power-relations between all participant groups. Although this would risk the course organizers dominating the early learning encounters rather than facilitating participants to develop their own hierarchy of relations, an early emphasis on these power relations could provide a space for more critical discussions of knowledge forms and customs throughout the course. Such discussion could also help to establish the necessary affective bonds and social ties between students and the grassroots communities that are needed to generate a more transgressive social learning process.

Aligned with the above was the discussion of power-relations and assumptions between researchers and participants in terms of research bias. Affirmation biases held by the co-researchers was discussed, evidenced by the focus group questions being centered uncritically around value-heavy concepts such as "diversity," "collaboration," and "empowerment." Furthermore, although we believe there was a high degree of researcher-subject trust present during the research, minimizing the social biases of participant assumptions about the meaning of questions and desirable answers, there was and always will be differences in hierarchy between researcher and subject. In retrospect, being more attentive and critical to these hierarchies could have opened up other unanticipated research results in terms of how we learn and relate to one another, especially if they form part of a wider conversation, for example, through dialogical networks of place-based learning communities [83].

### 5.5. Concluding Remarks

This paper has explored the levers and barriers to bridging forms of learning across the diverse contexts of higher education and community-based learning. Overall, results show that ICT, relations to place, and intercultural communication acted as levers towards bridging forms of learning between participants, facilitating critical reflection and the generation of affective relationships between participants. However, underlying power structures between participants acted as a barrier to learning, and needs future research so as for educational boundaries to be genuinely transgressed. In the case of this research, this could mean carrying out an assessment of power-relations between all participant groups, and more critically engaging in the unintended anticipatory learning resulting from transgressive encounters between participants. Overall, this suggests that as educators we need to be more anticipatory to emergence and inevitable element of uncertainty about where learning pathways are going.

Finally, based on the findings of this paper, further developing transgressive learning requires diversity and dissonance, in order to deepen the learning, recognize multiple ways of knowing and being in the world. Realizing such learning ecologies, however, means encountering strong structural barriers to bridging learning forms. This appreciates that although confronting inequities and power imbalances is important, it is a fragile process which requires responsible and responsive ethos to fostering both a renewal of higher education [34] and intercultural empathy.

**Author Contributions:** All authors helped conceive and design the research for this paper. The conceptualization, formal analysis and validation was carried out by authors T.M. (Thomas Macintyre), M.C., A.E.J.W. and V.C.T., while the methodology and investigation was carried out by T.M. (Thomas Macintyre), M.C., T.M. (Tatiana Monroy), M.O.Z. and T.V., with close supervision by A.E.J.W. and V.C.T. Original draft preparation, writing, review and editing was led by lead-author T.M. (Thomas Macintyre), with support from all co-authors. All authors have read and agreed to the published version of the manuscript.

**Funding:** This research was funded by the International Social Science Council TKN 150314115141; and COLCIENCIAS, the Colombian Administrative Department of Science, Technology and Innovation, through the Estancias Postdoctorales 784-2017, Codigo C145I000000000998-1.

**Conflicts of Interest:** The authors declare no conflict of interest. The funders had no role in the design of the study; in the collection, analyses, or interpretation of data; in the writing of the manuscript, or in the decision to publish the results.

**Appendix A**

Focus group questions. Questions originally asked in Spanish (see original questions below). The translation is the first author's interpretation. Information in brackets is for the interviewers use only.

1.   How did you experience the course? (A general and open question to start the conversation.]
2.   How did you experience the diversity of sources of traditional/ancestral, academic, and practical knowledge in the same course? What did you learn? Please provide an example. (Learning to know. Important to ask the necessary follow up questions to explore if the participants broke any barrier in their knowledge or way of knowing.]
3.   How did you experience the process of connecting with participants from other communities and backgrounds? Did this interaction help you learn? If so, how? Did this course in any way transform your way of understanding or experiencing the world? How? (Learning to be.)
4.   How did you experience the collaborative-practical process of the group work you were part of, and with the other participants, to generate tourism initiatives that address local challenges (workshops and videoconference)? To what extent has this course empowered you in your personal actions to generate transformations in your territories? (Learning to do.)
5.   Has there been anything that surprised you during this course; something you did not expect to learn? Is there anything else that you have learned apart from what we have already talked about? (Learning to anticipate.)
6.   What activity did you like best during the course? What activity do you not like? Which technological resource did you like the most, and which one did not? (Whatsapp, virtual learning platform, videoconference.)
7.   What was the biggest challenge you encountered during the course?
8.   The aim of this course was to "Strengthen the capacities, tools and knowledge of participants in the development of community and sustainable initiatives of tourism of origin, through the recognition of local knowledge, appropriate technologies and organizational knowledge that are exchanged through the use of ICT for the generation of a specific product." Do you believe that this goal was achieved? Why?
9.   Do you have any suggestions to improve the course in such a way that this objective is achieved and your learning process is maximized?

Original questions in Spanish:

1.   ¿Cómo experimentaron el curso? [pregunta general abierta como iniciador]
2.   ¿Cómo experimentaron la diversidad de fuentes de conocimiento tradicional/ancestral, académica y práctica en el mismo curso? ¿Qué aprendiste, dame un ejemplo? [Sobre aprender a conocer, importante hacer las preguntas necesarias de más para saber si los participantes rompieron alguna barrera en su conocimiento o forma de conocer]
3.   ¿Cómo experimentaron el proceso de conectarse con otros participantes de otras comunidades y antecedentes? ¿Esta interacción les ayudó a aprender, cómo? ¿Este curso de alguna forma transformó su forma de entender o experimentar el mundo?, cómo? [Sobre aprender a ser]
4.   ¿Cómo experimentaron el proceso colaborativo-práctico de trabajo en grupo entre ustedes, y con los otros participantes, para generar las iniciativas de turismo que abordan los desafíos locales (convite y videoconferencia)? ¿Hasta qué punto este curso lo ha empoderado en sus acciones personales para generar transformaciones en sus territorios? [Sobre aprender a hacer]
5.   ¿Ha habido algo que lo ha sorprendido en este curso, de lo que no esperaba aprender?¿Hay algo más que haya usted aprendido a parte de lo que ya hemos hablado?

6.　　¿Qué actividad le gustó más del curso? ¿Qué actividad no le gusto? ¿Qué recurso tecnológico le gusto más y cuál no? [whatsapp, plataforma virtual, videoconferencias]

7.　　¿Cuál fue el mayor desafío que encontró durante el curso?

8.　　Este curso KORU tenía como objetivo: Fortalecer la conexión entre Sociedad, investigación y tecnología a través del reconocimiento del valor de conocimiento indígena local y la participación activa de diversos actores para abordar un desafío local común. ¿Ustedes creen que se alcanzó este objetivo? Porque?

9.　　¿Tiene alguna sugerencia para mejorar el curso de tal forma que se alcance este objetivo y se maximice su proceso de aprendizaje?

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
