# Peer review of "Transgressing Boundaries between Community Learning and Higher Education: Levers and Barriers"

_sustainability, doi:10.3390/su12072601_

Round 1

Reviewer 1 Report

Thank you for the opportunity to review the manuscript “Transgressing educational boundaries: Levers and barriers towards bridging community learning and higher education through the Koru educational approach.”

The paper is a comprehensive description of a learning program connecting HE and communities. It is within the scope of Sustainability, and reads well. I suggest the following issues need to be addressed before considering the manuscript for publication:

·      Abstraction: the manuscript is very rich in personal detail and reflection. While this underlines the relational approach taken by the authors, the discussion should take the findings to a more abstract level in relation to the conceptual background and the literature.

·      Conversation: while the manuscript is strongly linked to the discourse on transgressive learning and the T-learning project, it does not sufficiently engage in conversation with other work that has addressed similar questions—although maybe using a different terminology (transdisciplinary, transformative etc.). Some examples: work by Charles Francis, Geir Lieblein, Anderson et al. 2018, Akkerman 2011, Probst et al. 2019, Molderez 2017.

·      Focus: the richness in detail is a strength of the paper. However, it is part of the systemic disfunction of academia that we don’t have time to read—considering the importance of the subject, I suggest to substantially condense the narrative and focus on the main points.

·      Clarity: in the manuscript, the boundaries between what you did, what you learned and what this means are not sufficiently clear. Working on this distinction would make it easier for others to learn from this case. Since Koru was not developed with the course participants, I suggest that it is presented as part of the methodology.

Detailed observations:

Language

In general, the manuscript reads well. Some sections (e.g. Chapter 2, 4.2.1) need language revision. Some typos remain throughout the manuscript.

Title

I find the title a bit woolly. First, transgressing and bridging have similar meanings. Second, and to my knowledge, you can only bridge something (a gap, or a canyon) that separates two things-such as the boundary between community learning and HE. Third, most of the readers will not be familiar with the Koru approach, and I don’t think Koru is at the heart of the manuscript’s message. Some suggestions:

“Transgressing boundaries between community learning and higher education: levers and barriers.”

“ICT, relations to place and experience as levers for transgressive learning: a case study from Colombia.” 

Abstract

The abstract is a succinct summary of the study. The reference to the SDGs reads a bit artificial – as a justification, I would find something like “locally relevant responses to sustainability challenges” more convincing.

l. 18: see above on title – bridge education with learning environments…

1.     Introduction

l. 39-50: I enjoyed the emotional and personal jump start – but suggest to remove the section to save space.

l. 52-59 some references would be helpful, illustrating the discourse in Colombia.

l. 58: Who is “we”?

l. 67: “easy feat” – I find that the frequent use of judgemental statements weakens the narrative of the article. I suggest to scan the manuscript and remove such statements and adjectives where possible.

l.67ff the need for transgressive learning is introduced here and taken up again in Chapter 2. I suggest to move everything that justifies your study from Chapter 2 to the Introduction, and revise Chapter 2 towards a more specific analytical framework.

l. 78-89 Suggest to delete – instead, I would like to know more about the initiatives that took part in the activity, probably either in the introduction or the methods section.

Currently, the description of the initiatives pops up here and there – but lacks detail. Who are the participants? How and where do they live? What do you know about the challenges they face and what they value? Figure 1 helps, but is not sufficient. Maybe a table introducing each initiative plus the other participants along a set of dimensions would work.

l. 99 Multi-stakeholder: I don’t think you can claim that the course is a multi-stakeholder project – this would have required that you mobilize all actors who have a stake in tourism-related issues in the three communities. Maybe community learning or similar would fit better. This applies to the whole document.

l. 107ff: detrimental effects of mass-tourism: can you provide some evidence here? Studies in other parts of the world found that local communities preferred the separated world of commercial mass tourism over the typically shoestring backpacking type.

l. 115 – 118: During the development of the course, an approach emerged, and the course was then the application of the approach?

For clarity, I suggest to convincingly establish the need for transgressive learning here, and thus develop your objectives on a general level.

            -> Developing a general transgressive learning model.

            -> Apply the model and learn from its application.

In general, throughout the manuscript, the research/learning process which led to Koru and Turismo de Origin should be described in a more digested fashion. I assume the reality of the process was indeed organic and adaptive, but it is hard for the reader to connect the bits and pieces. Considering that Koru was designed by the action research team, I would rather see it as part of the methodology.

l. 124ff. The sections of the manuscript should follow a logical flow, so that this kind of reading instruction is not necessary. Remove.

2.     Conceptual framework

l. 140ff: This is an interesting statement – but out of place. It is not related to the conceptual framework, but part of the process.

l.154ff. I suggest to move all conceptual considerations that justify your work to the introduction. Chapter 2 could then specify how you frame your analysis (what you later refer to as thematic codes).

l.158 Particularly in the Latin American context, some reference to Freirean emancipatory pedagogies and how they relate to transgressive learning would be appropriate.

l.162: Agenda 2030?

l. 240-248 Redundant. Introducing once why you do the study and ask the research question should do (introduction).

3.     Methodology

l. 257ff You should clearly justify the design of a new approach in the introduction, and how it looked like in the methods.

Similarly, it is not clear at this point how Turismo de Origin evolved. In the context of transgressive learning, I would also expect that you discuss how you addressed the paradox of initiating a process while trying not to dominate it (Habermas, Theory and Practice). More information on how and why the facilitators worked with the communities would be helpful.

l. 277ff I believe this is the first time you clearly mention this first objective. It should be established earlier. See comments above for having a clear storyline.

l. 289ff and Appendix A: the empirical design is a weakness of the paper. First, you should make clearer who participated in the course, and I would like to know more about the initiatives: why do they exist? Who is represented? What is their identity and which values do they share? Why did the facilitators approach/initiate these groups?

Second, it is unfortunate that many of the focus group questions have leading elements that may have contributed to social desirability biases – e.g. “diversity”, “connecting”, “collaborative-practical”, “empowered”, “surprised”. While it is obviously too late to improve this, I would expect that the researchers discuss these limitations and their own expectations and biases.

The several rounds of coding are well described. It would be helpful to have the four forms of learning and the related FGD questions in a table, as an empirical analytical framework.

4.     Results

The general part of the Koru design should be presented in the methodology. In the results chapter, you should focus on the findings along the analytical framework.

In general, the chapter contains many methodological aspects (e.g. l. 329-334) and interpretations (e.g. “This has the advantage of working with communities where trust is already developing, but has a potential disadvantage of limiting the overall objectives of the course to the contexts of the communities.” L. 370ff.).

Methodology, results and discussion needs to be disentangled. Move all design aspects to the methodology.

Figure 2: A spiral is nice, but the logic of the process is linear, as confirmed by the numbering. If this is true (or are there any ladders in the spiral that are not shown?), then a classic linear presentation would be easier for the reader to grasp.

l.367 From my experience, I assume that the trust established between initiatives and facilitators is key for the learning process. This element, and the perception of the participants that the facilitator actually cares, is probably key in a relational learning approach. I think this should be discussed in more detail.

While the relationship to the community initiatives seem to have evolved from the parental project, it is not clear how the student group was mobilized. Why Social Work students? How do they relate to issues of tourism in rural areas? Would you have worked with another student group if the main challenges identified by the communities would have been agronomic? I am definitely not a proponent of expertism, but to balance the legitimacy of different epistemologies (as you discuss in l. 509ff and 863ff), this is a relevant issue.

Sections 4.2.1 to 4.2.5 – I enjoyed reading these notes – they would make excellent blog posts. For the paper and as mentioned above, I suggest to focus on the digested results along the dimensions to know, to be, to make change, to anticipate (each being a subsection). This would help the reader to connect conceptual framework, methodology and results.

The levers and barriers, in turn, should be the focus of the discussion chapter.

The quotations read very authentic and reflect the excellent job the facilitators have done in enabling critical reflection. I would like to have more information on who is talking – e.g. is Alumni student Alejandra part of the student group? Alumni of?

5.     Discussion

The major findings are well emphasized, but not sufficiently put into perspective.

l. 822-831 Redundant.

Remove all specific results and quotations. The discussion should abstract the findings to a general level.

l. 852ff Connection between communities and students – beyond power relations and a lack of communication, you may want to consider positive distinctiveness among the community members (confirming a positive self-concept) and related othering (what do the bookworms know) as factors.

The discussion is still very close to the empirical experience of the work – further engaging in a conversation with earlier work would help the reader to see the general relevance of this study.

Concluding remarks

The concluding remarks should be rewritten. You mainly discuss Koru as a type of curriculum, but this is not the main point of the study.

As a conclusion, I suggest to re-state the objective and summarize the major arguments of the study. Key steps in further developing transgressive learning could be pointed to, based on the findings of this study.

Author Response

Dear reviewer,

Thank you very much for taking the time to read the manuscript in depth, and providing constructive criticism. We have thought a lot about your suggestions, and the changes we have made based on your comments have been invaluable for improving the manuscript. In the attached file are your comments with the changes addressed. There were too many changes made to use track changes, as this would have distorted the document too much, but we have added line numbers where appropriate in the table.

We hope you find the manuscript improved and enjoyable.

Reviewer 2 Report

I really enjoyed reading the paper and learning from it. But one thing that strikes me is that this approach is actually a variation of what we´ve been doing in Brazil since the 1980´s. We work in an environmental education that is grounded in three major domains: knowledge, participation and ethics. So often when I read about ESD it always seems to me that researchers in this field are “discovering” something we´ve known all along. The work by SA researchers is very close to what we do. Maybe it´s because of social economic similarities between our countries. So although I do appreciate and respect this TL approach, I´d like to see more references to the Brazilian work in the EE field. 

Author Response

Dear reviewer,

Thank you for taking the time to review our manuscript. Yes, the TL approach owes a lot to the work of Latin American scholars such as those in Brazil in the field of EE, building on many of their ideas, such as critical ecology. We have added further references to do justice to this, for example:

Freire, P. Pedagogy of the Oppressed; Continuum: New York, 1970; p. 125;.

Thiemann, F.T.; de Carvalho, L.M.; Torres de Oliveira, H. Environmental education research in Brazil. Environ. Educ. Res. 2018, 24, 1441–1446.

de Sousa Santos, B. Beyond Abyssal Thinking: From Global Lines to Ecologies of Knowledges. Rev. Fed. Am. Health Syst. 2007, 30, 45–89.

de Sousa Santos, B. Epistemologies of the South: Justice against epistemicide; Routledge: London and New York, 2016;.

Souza, D.T.; Wals, A.E.J.; Jacobi, P.R. Learning-based transformations towards sustainability: a relational approach based on Humberto Maturana and Paulo Freire. Environ. Educ. Res. 2019, 1–15.

A relevant part in the conceptual framework reads: 

"Decolonizing pedagogies therefore refer to pedagogies which promote marginalized forms of knowledge, such as indigenous and local knowledge (ILK), which has a strong tradition in Latin America in line with Freirean emancipatory pedagogies and environmental education in Brazil [19–22], alongside African movements [16,17]. Line 150

We hope you are satisfied with these changes. 

Kind regards, the author team

Round 2

Reviewer 1 Report

The manuscript has improved substantially, and the flow of arguments is much clearer. Some issues remain, that I suggest need to be addressed before publication:

Implicit, but obvious normative orientation of the authors: I can relate to the challenge of passionately being part of a process, while being asked to critically reflect on the experience. Currently, the tension between inspired activism and critical analysis is not explicitly addressed—this will make it easy to dismiss the work as biased/activist. Resonating with the personal and relational spirit of the paper, I suggest to add a paragraph to the introduction making this challenge explicit. Conversation beyond the transgressive community: while a few references have been added, the manuscript still does not reflect on the main findings (ICT, context, transcultural) in relation to earlier work from different paradigmatic communities.

Detailed observations:

Introduction

l.55: the UN is a multi-party body with no agency to make something clear. I suggest to replace with: “As made clear in the document XY, ”

l.55-71 You are oscillating between different understandings of education for/as sustainability, but these different understandings have fundamentally different implications regarding the pathway towards a more sustainable future. The justification in line 58 is very instrumentalist, while Wals’ understanding (as I interpret it) would see education as sustainability. In line 62ff, your understanding of education is very instrumentalist, and does not connect well with the thinking of reference 5. In line 68, you refer to “education for sustainable development”—the concept is again different from education for/as sustainability, and is critically discussed (see Hellberg and Knutsson 2018). Indeed, they see ESD as a potentially colonizing endeavour. Also, your definition of a “sustainable world” is not clear, but “colonial and neoliberal practices” seem to be a kind of antonym. I am also worried about the idea that transgression relates only to colonial and neoliberal  practices: in my understanding, transgression is about co-learning, cognitive justice, and agency across established boundaries. Prescribing values and norms would be in contradiction to learning as emancipatory process.

I suggest to  clearly state your understanding of education for/as sustainability, and justify your approach accordingly throughout.

l.82 I have limited experience in Colombia, but my experience in other parts of the world makes me doubt this statement: I assume it is rather the absence of effective regulation which leads to a kind of rogue capitalism, than actually implemented policies.

l.110 Who are the organizing teams? Not clear at this stage.

l.114-118 Please crosscheck that your references 9 and 10 really discuss tourism in Colombia. 

Conceptual framework

l.141 Check reference 18. IPCC is not the author of the report according to your ref list. Also, I cannot find any reference to praxis (this would be quite remarkable) or theories of decolonization in the report.

l.170ff This is repetitive. You discuss the necessary qualities of education before. This paragraph has no theoretical implications for your study.

l.187ff Head-heart-hands should acknowledge Sipos et al. (2007)

Methodological background

l.223-239 Suggest to shorten or integrate. The evolution of the approach is not discussed later, so seems to have limited relevance for your findings.

Results

In several sections of the results chapter, you interpret and reflect on the findings – e.g. l532-535, l547-549, l56, l582-583, l594-598, l619-622, l640. All interpretation should move to the discussion.

L640 “This result…” Confusing – absence resulted in lack of learning or unintended learning as a form of learning to anticipate?

Discussion

L666-689 I find the reference to SDG4 artificial. In fact, SDG4 and its targets make no reference to transformation or transgression – the focus is on (a more instrumental understanding of) knowledge and skills (Target 4.7).

The whole section reads like a justification of your research – which by now should be well justified. Remove or integrate into the introduction.

L721 “with the video-conference…” Rather a result.

L760-765 Result.

L822ff The discussion of the limitations is helpful. As mentioned in my first review, and in the spirit of critical self-reflection, I would also expect that the authors address the paradox of initiating a process while trying not to dominate it. 

L830ff “socially desirable biases” – there might be biases that are socially desirable, but what you are describing well here is the confirmation bias (looking for what you want to find). Considering your research instrument, you should discuss in addition the social desirability bias, which occurs when respondents make assumptions about the meaning of questions and desirable answers.

Round 3

Reviewer 1 Report

The manuscript has improved, and I suggest to accept pending a final check of spelling (e.g. l.56) and grammar (e.g. inconsistent use of commas).

Happy new year and all the best for further endeavours in transgressive learning!

Author Response

Dear reviewer,

Thank you for all your effort in reviewing our manuscript and providing constructive comments. We have made a last grammar revision (see document with track changes), and feel any further grammatical changes can be made by the final revision by the journal.

We hope you are enjoying the new year!
